# Collection of economic data using UB-04s: Is it worth the effort? Evidence from two clinical trials

Lucas Higuera[1][*], Eleni Ismyrloglou[2], Xiaoxiao Lu[1][¤], Jennifer Hinnenthal[1], Reece Holbrook[1]

1 Medtronic Inc, Cardiac Rhythm & Heart Failure, Mounds View, Minnesota, United States of America,
2 Medtronic Bakken Research Center B.V, Maastricht, Netherlands

☯ These authors contributed equally to this work.
¤ Current address: Janssen Scientific Affairs LLC, Titusville, Pennsylvania, United States of America
* lucas.f.higuera@medtronic.com

**Data Availability Statement:** Complete data cannot be shared publicly because of confidentiality restrictions of clinical trials (Micra IDE and WRAP-IT trials) and access restrictions to

## Abstract

Cost collection using UB-04 forms for economic evaluation is challenging, as UB-04 collection is time and effort intensive and compliance is imperfect. Alternative data sources could overcome those challenges. The objective of this study is to determine the usefulness of UB-04 data in estimating hospital costs compared to clinical case report form (CRF) data. Health care utilization costs were compared from financial information in UB-04s and from an assignment process using CRF data, from the WRAP-IT (23 infections) and the Micra IDE trials (61 adverse events and 108 implants). Charge-based costs were calculated by multiplying charges in UB-04s and hospital-specific Cost-to-Charge ratios from the Centers for Medicare and Medicaid Services cost reports. The cost assignment process used clinical information to find comparable encounters in real world data and assigned an average cost. Paired difference tests evaluated whether both methods yield similar results. The mean difference in total infection related costs between methods in the WRAP-IT trial was $152 +/-$22,565. In the Micra IDE trial, the mean difference in total adverse event related costs between methods was -$355 +/-$8,298 while the mean difference in total implant related costs between methods was $-3,488 +/-$13,859. Wilcoxon tests and generalized linear models could not reject the difference in costs between methods in the first two cases. Cost assignment methods achieve results similar to costs obtained through UB-04s, without the additional investment in time and effort. The use of UB-04 information for services that are not mature in a health care system may present unexpected challenges, necessitating a tradeoff with other methods of cost assignment.

## Introduction

When evaluating the success of a medical device, the primary concerns are typically safety, efficacy, and cost-effectiveness. Cost-effectiveness, which can be determined by value analysis, is critical in determining the value for money a new treatment or technology brings to

administrative claims data (Medicare claims) and cost data (Premier Hospital Database). Minimal underlying data sets are included in the Supporting information files.

**Funding:** LH, EI, JH, and RH are all employees of Medtronic plc. XL is a former employee of Medtronic plc.

**Competing interests:** This study was funded by Medtronic, as well as the clinical trials used in it; this does not alter our adherence to PLOS ONE policies on sharing data and materials. The publication of study results was not contingent on Medtronic's approval or censorship of the manuscript. All authors, except for Dr. Lu, currently receive wages and benefits from Medtronic, and are minority stockholders of Medtronic. Dr. Lu received wages and benefits from Medtronic as a former employee.

consumers, providers, and payers of health care. One way to enable value assessment is to collect health care utilization data alongside clinical trials. This type of evaluation, referred to as "piggyback" economic evaluation [1], allows for timely development of economic evidence. Other approaches may consist of retrospective analysis of administrative or claims data, including chart review or modeling. Regardless of the approach, these data help decision makers such as payers and providers to evaluate the potential economic implications of a therapy in addition to its primary clinical impact.

In the United States (US), the options for collecting trial participants' health care utilization during a study often include filling out case report forms (CRFs) or obtaining the Form CMS-1450, more commonly known as the UB-04 form (S1 Fig). The UB-04 form is the standard claim form used by institutional providers to bill Medicare Administrative Contractors and is widely accepted by all insurance carriers. While collecting UB-04s provides a uniform way to gather information on hospital charges, it only accounts for hospital-based services, requires conversion to estimate cost and is applicable only to the US. Further, sites may not want to share charge data with the study sponsor. Collecting UB-04s can be expensive and time consuming and may also prove difficult given that a site's research coordinator is separate from the billing clerk who may otherwise not be involved in or trained to the clinical trial protocol. Despite potential difficulties, UB-04s may be the only way to get timely collection of economic data if the procedure or indication of interest is not yet included in the existing procedure and diagnosis codes, and therefore will not appear in alternative costing data sources. An additional advantage of UB-04s is that it provides an accurate estimate of the costs incurred within the trial.

Alternatively, CRFs can capture health care utilization (HCU) along with procedures and diagnoses of interest; yet come at the expense of no direct cost data and potential lack of detailed care provided during an encounter. Waiting out the typical lag in administrative or claims data, while an option, would otherwise leave economic analyses delayed for a year or more. How then do study sponsors decide whether to collect economic data via CRFs, UB-04s or administrative data? The International Society for Pharmacoeconomics and Outcomes Research (ISPOR) Good Research Practices Task Force Report [2] is one resource providing guidance about decision criteria for collecting economic data. This report states the possible trade-offs across accuracy, feasibility, generalizability, and cost among costing methods, but the report doesn't provide specific comparisons across costing methods. In this study we take advantage of two clinical trials where UB-04 data were collected, assess the feasibility of calculating costs with information from UB-04s and CRFs, and compare the results of both approaches. This study aims to equip the reader with qualitative considerations for collecting cost data, as well as provide real world examples of quantitative differences in cost estimates based on two different approaches.

## Methods

### Data

This study uses data from two different clinical studies that collected UB-04s per the study protocol. First, the World-wide Randomized Antibiotic EnveloPe Infection PrevenTion Trial (WRAP-IT) was a post-market, interventional clinical study that collected information between 2015 and 2018 related to Cardiac Implantable Electronic Device (CIED) infection complications 12 months after implantation [3]. Second, the Micra Investigational Device Exemption (IDE) study was an interventional study that collected information on implants of Micra™ (Medtronic plc, Mounds View, MN, US), a leadless intracardiac transcatheter pacing system, and on adverse events 6 months after implantation between 2015 and 2017 [4]. Even

though both studies enrolled patients worldwide, we used information only for patients enrolled in the US for whom UB-04s were available.

## Cost comparison

For each study we compared HCU costs from two sources: costs obtained from financial information in UB-04s (UB-04 charge-based costs), and costs obtained from an assignment process that uses clinical information in these forms to find average costs from equivalent health care encounters (assignment-based costs). UB-04 charges-based costs were calculated by multiplying charges in UB-04s and hospital-specific inpatient or outpatient Cost-to-Charge ratios (CCRs) from the Centers for Medicare and Medicaid Services' (CMS) Healthcare Cost Report Information System (HCRIS) from the year of the HCU [5,6]. The objective of this study is not to compare costs by treatment status within each trial, but to compare total costs by costing method within each trial.

The cost assignment process uses diagnosis, procedure codes and other clinical information (place of service, type of medical device implanted, type of utilization, etc.) listed in the UB-04s of each HCU to find comparable encounters in real-world data and assign an average provider cost of these encounters to the observed utilization. Provider cost data comes from the Premier Hospital Database (PHD), a data source from Premier Applied Sciences® that covers over 131 million inpatient visits, representing approximately one in every five inpatient discharges in the US. Hospitals included in the database are a national representation in terms of regional distribution, urban vs rural hospital, teaching vs non-teaching institutions, and hospital bed size [7]. Hospitals and healthcare systems submit health care utilization and financial data from inpatient and outpatient encounters. PHD calculates provider costs either directly from the provider accounting system or by transforming provider charges into costs using Medicare CCRs. All assignment-based costs were adjusted to 2017 (WRAP-IT CIED infections, to be consistent with previous economic analyses of the WRAP-IT trial [8]) or 2018 (Micra IDE implants and adverse events) US dollars using the Consumer Price Index for Medical Care published by the US Bureau of Labor Statistics.

## WRAP-IT cost comparison

The cost assignment process for the WRAP-IT CIED infections looked for inpatient, hospital outpatient, and emergency department utilizations separately in the PHD. Inpatient utilizations were matched by year, Diagnosis-Related Groups (DRGs), type of CIED, and type of procedure performed (CIED removed and replaced, CIED removed and not replaced, new CIED implanted, and no CIED intervention). Only encounters with an infection diagnosis code (S1 Table) in the matching data were selected for the assignment process. An average cost per day of stay was computed in PHD and later multiplied by the observed length of stay to calculate total assignment-based costs. Hospital outpatient utilizations were matched by year, a subset of procedure and diagnosis codes, and type of encounter (same day surgery or observation stay). Emergency department visits were matched by year, a subset of diagnosis codes, and attending physician specialty (primary care or emergency medicine). The average costs in these last categories were assigned without adjustments for length of stay. Procedure and diagnosis codes were selected based on relevance to cardiovascular diseases and having a resulting matched sample size larger than 2; S1 Table lists all codes used in the match.

## Micra cost comparison

Leadless pacemakers entered the US market in 2017, but the Micra IDE study implants occurred between 2014 and 2015. This poses two complications for the cost assignment

process: first, few Micra implants appear in the PHD in 2014 and 2015; and second, procedure codes and DRGs in the Micra IDE study UB-04s don't reflect the actual reimbursement procedures used for Micra implants. We adjusted the cost assignment process for the Micra IDE study implants by 1) reassigning the DRG codes in the UB-04s into the DRG codes that current payment practices use to reimburse Micra implants (DRGs 228 and 229) based on the existence of major complications, and 2) using PHD encounters in 2018 in the cost assignment process for inpatient and outpatient Micra IDE study implants. Inpatient Micra implants were matched on the newly coded DRGs, and an infection indicator based on an infection diagnosis code (S1 Table). An average cost per day of stay was computed and multiplied by the observed length of stay to obtain total assignment-based costs. The assignment-based costs for all outpatient Micra IDE study implants were based on a single average cost of outpatient Micra implants in the PHD in 2018.

The cost assignment process for the Micra IDE study adverse events matched inpatient utilizations by year, DRG codes, and a subset of procedure codes. Average costs per day were then calculated and multiplied by the observed length of stay to obtain total assignment-based costs. Outpatient and Emergency department utilizations were matched by year and a subset of procedure codes. S1 Table lists all codes used in the match.

## Statistical methods

For each case, we plotted box plots of costs per costing method and for the difference between costs, and we graphed scatter plots with charge-based costs in the x-axis and assignment-based costs in the y-axis and estimated a best fit line per plot. We used Wilcoxon tests to non-parametrically examine differences in the distribution between costing methods. Since health care costs data usually follow right-skewed distributions, traditional t-tests are not recommended. In order to test the differences by costing method, we estimate a generalized linear mixed model (GLMM) with a log link and a gamma family using costs as the dependent variable, a dichotomous indicator for costing method, and patient random effects. If the coefficient of the costing method is not statistically different than zero, then the costing methods are not different at the mean. Additionally, to account for potential differences in coding methods by trial site, we estimate a GLMM with patient and site cross-random effects.

## Results

### WRAP-IT CIED infections

There were 70 major CIED infections observed in the WRAP-IT trial. Among these, 43 infections were at sites in the US and eligible to collect UB-04s, and a full set of UB-04s for all related HCUs were collected for 23 infections (53.5% of US infections). The primary reasons for missing UB-04s for infections were hospitalizations that occurred separate from the investigational site and research coordinator unfamiliarity with hospital billing systems.

For the infections with a full set of UB-04s, the mean overall cost of infection events was $48,848 +/- $40,272 using the UB-04 charge-based cost method and $49,000 +/- $28,130 using the assignment-based cost method (Table 1; Fig 1 Panel A). The mean difference in total infection related costs between methods was $152 +/- $22,565. The paired cost estimates are plotted in Fig 2, Panel A with assignment-based costs on the Y axis and UB-04 charge-based costs on the X axis. The least squares regression line had a slope of 0.59 with a correlation coefficient of 0.84. Results from the Wilcoxon tests and GLMM (bottom of Table 1) cannot reject the hypothesis of equal results from both cost assignment methods.

**Table 1. Summary statistics and hypotheses tests.**

| | WRAP-IT CIED infections | | | Micra IDE Adverse events | | | Micra IDE Implants | | | | |
|---|---|---|---|---|---|---|---|---|---|---|---|
| | Charges-based | Assign.-based | Difference | Charges-based | Assign.-based | Difference | Charges-based | Assign.-based | Difference | Var. Assign.-based | Difference with Var. |
| N | 23 | 23 | 23 | 66 | 61 | 61 | 108 | 104 | 104 | 104 | 104 |
| Mean | 48,848 | 49,000 | -152 | 12,095 | 12,529 | -355 | 21,003 | 24,663 | -3,488 | 23,491 | -2,316 |
| SD | 40,272 | 28,130 | 22,565 | 16,880 | 14,187 | 8,298 | 18,495 | 17,365 | 13,859 | 13,593 | 12,937 |
| p05 | 5,612 | 10,293 | -49,187 | 1,610 | 1,458 | -7,589 | 8,237 | 12,555 | -25,668 | 14,530 | -17,414 |
| p25 | 22,414 | 31,665 | -7,523 | 4,221 | 5,643 | -3,692 | 12,755 | 14,530 | -6,832 | 14,530 | -6,714 |
| p50 | 35,253 | 47,070 | -83 | 8,981 | 9,557 | -1,482 | 16,853 | 14,530 | -2,806 | 15,576 | -3,136 |
| p75 | 63,555 | 63,656 | 17,257 | 14,984 | 13,918 | 106 | 21,633 | 28,132 | 2,155 | 27,719 | 396 |
| p95 | 141,300 | 109,299 | 34,681 | 34,613 | 25,115 | 13,621 | 51,203 | 50,637 | 11,612 | 45,438 | 11,672 |
| *Wilcoxon* | | | | | | | | | | | |
| Sign-rank Z Stat. | | 0.061 | | | -2.331 | | | -3.65 | | -4.53 | |
| Sign-rank p-value | | 0.9515 | | | 0.0198 | | | 0.0003 | | <0.001 | |
| Sign-test two-sided prob. | | 1.0000 | | | 0.0044 | | | 0.0005 | | <0.001 | |
| *GLMM* | | | | | | | | | | | |
| Coefficient | | -0.1098 | | | -0.1125 | | | -0.1650 | | -0.1632 | |
| Coefficient p-value | | 0.491 | | | 0.229 | | | <0.001 | | <0.001 | |
| Difference | | -5,429.11 | | | -1,375.38 | | | -3,674.41 | | -3,562.57 | |
| Difference p-value | | 0.495 | | | 0.235 | | | <0.001 | | <0.001 | |
| *GLMM with site variation* | | | | | | | | | | | |
| Coefficient | | -0.2675 | | | -0.1125 | | | -0.1832 | | -0.1780 | |
| Coefficient p-value | | 0.182 | | | 0.229 | | | 0.002 | | 0.002 | |
| Difference | | -13,875.09 | | | -1,375.36 | | | -4,131.40 | | -3,945.75 | |
| Difference p-value | | 0.197 | | | 0.235 | | | 0.002 | | 0.002 | |

GLMM: Generalized Linear Mixed Model, SD: Standard Deviation.

## Micra adverse events

There were 1,121 adverse events observed in the Micra IDE trial. Among these, 570 were at sites in the US and eligible to collect UB-04s, and UB-04s were collected for 61 adverse events (11.1% of US adverse events).

For the adverse events with UB-04s, the mean overall cost of adverse events was $12,095 +/- $16,880 using the UB-04 charge-based cost method and $12,529 +/- $14,187 using the assignment-based cost method (Table 1; Fig 1 Panel B). The mean difference in total adverse event related costs between methods was -$355 +/- $8,298 for those 61 adverse events with both cost estimates. The cost estimates are plotted in Fig 2, Panel B with assignment-based costs on the Y axis and UB-04 charge-based costs on the X axis. The least squares regression line had a slope of 0.72 with a correlation coefficient of 0.88. Here, the Wilcoxon tests reject the hypothesis of similar distributions (Table 1), but the GLMM cannot reject the hypothesis of equal means.

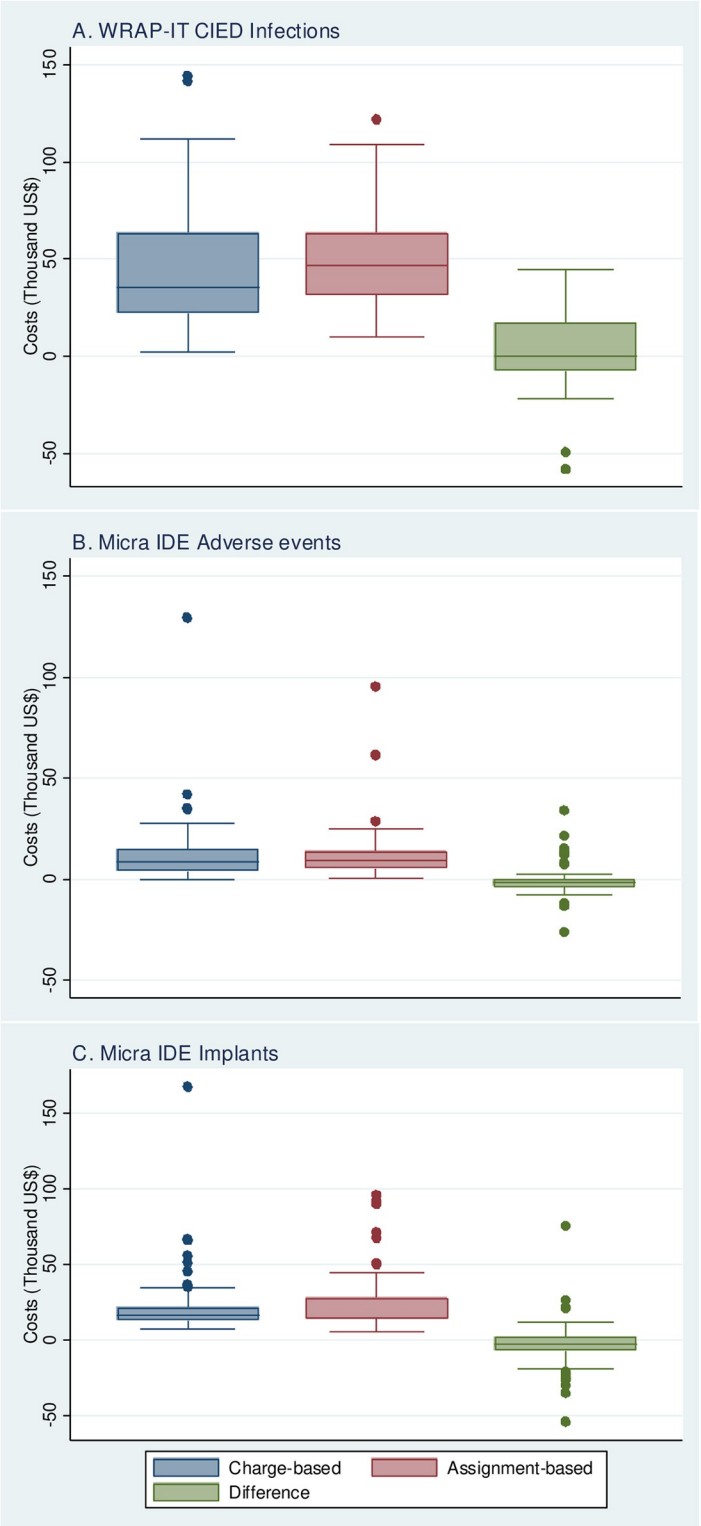

**Fig 1. Box and whisker plots—Charge- and assignment-based costs.** Box and whisker plots for cost (in Thousand US $) are shown from A) WRAP-IT CIED Infections, B) Micra IDE Adverse Events and C) Micra IDE Implants. Charge-based costs are shown in blue, assignment-based costs are shown in red and the difference between the two is shown in green.

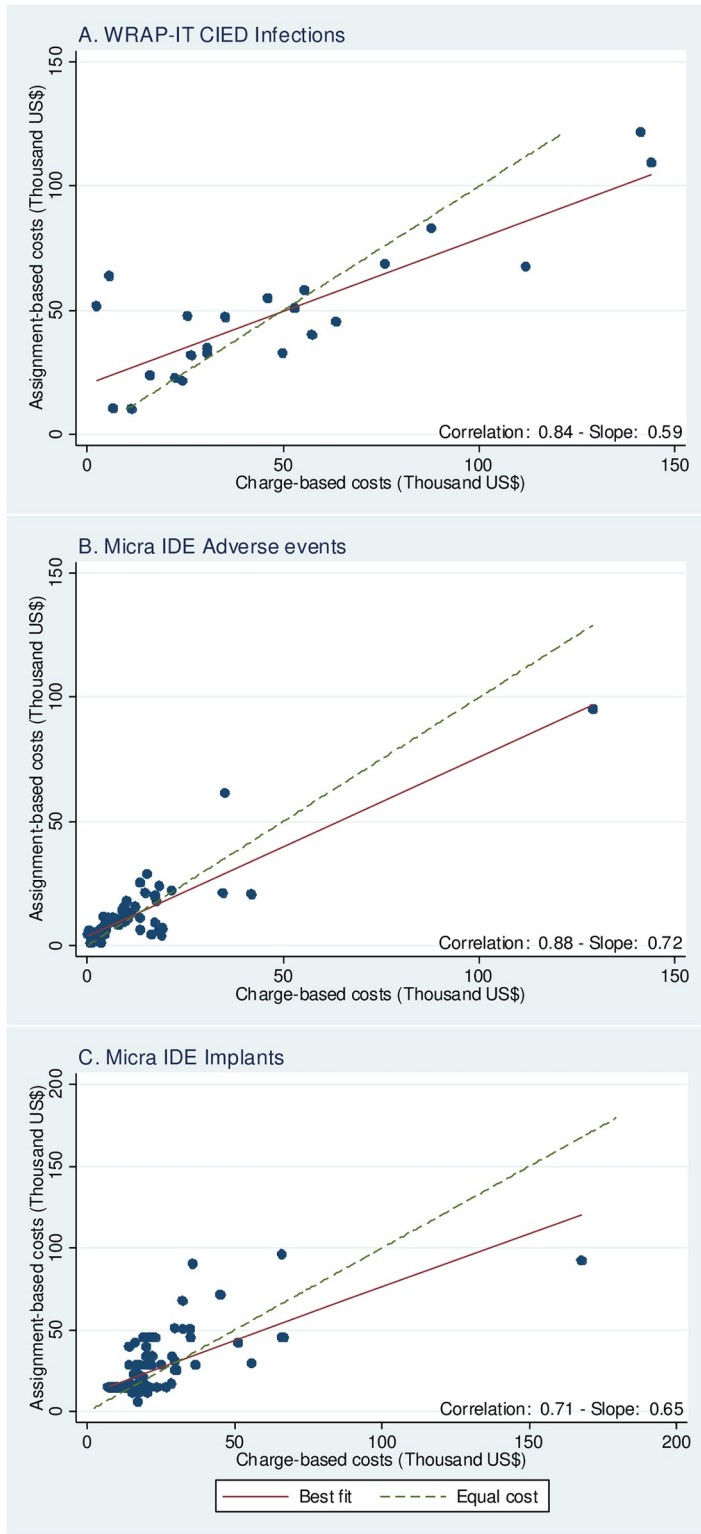

**Fig 2. Scatterplots of charge-based and assignment-based costs.** Scatterplots showing Charge-based costs (x-axis) vs Assignment bases costs (y-axis) are shown for A) WRAP-IT CIED Infections, B) Micra IDE Adverse Events and C) Micra IDE Implants. The best fit line is denoted with a solid red line and equal cost is shown with a dotted green line.

## Micra implants

There were 726 implant attempts observed in the global Micra IDE trial. Among these, 284 implants were at sites in the US and eligible to collect UB-04s, and UB-04s were collected for 108 implants (38% of US implants).

For the implants with UB-04s, the mean overall cost of implants was $21,003 +/- $18,045 using the UB-04 charge-based cost method and $24,663 +/- $17,365 using the assignment-based cost method. The mean difference in total implant related costs between methods was $-3,488 +/- $13,859 (Table 1; Fig 1 Panel C) for those 104 implants with both cost estimates. The cost estimates are plotted in Fig 2, Panel C with assignment-based costs on the Y axis and UB-04 charge-based costs on the X axis. The least squares regression line had a slope of 0.65 with a correlation coefficient of 0.77. Both Wilcoxon tests and GLMM reject the hypothesis of both methods yielding similar results (Table 1).

There were 16/28 sites (57%) that showed evidence of anomalous implant cost estimates, having had total charge-based cost estimates less than the expected purchase price of the Micra device alone. In this case, we implemented an alternative assignment-based (variable assignment-based costs) costing method, where we subtracted the cost of a Micra device from each matched encounter found in the PHD and recalculated the average cost per day of stay. We multiplied this number by the observed length of stay and then added back the cost of a Micra device to the total cost. The last two columns on Table 1 shows that the overall cost of implants using the variable assignment-based method was $23,491 +/- $13,593, which reduced the difference with the charges-based costs to -2,136 +/- $12,937. Fig 3 shows the scatterplot of both costing methods; with respect to the normal assignment-based method, the correlation increased marginally, and the slope of the best fit line decreased. However, both the Wilcoxon tests and the GLMM still reject the hypothesis of similar distributions and equal means.

## Discussion

Our results show that the two costing methods analyzed (costs obtained through UB-04s and Cost-to-Charges Ratios, and costs obtained through CRF information and data assignment from administrative claims) estimate costs not statistically different from each other in two clinical trials with HCU established in the coding systems. In the case where the events analyzed (Micra implants) were still novel in the coding system, the cost methods yield results statistically different from each other. These differences are not derived from different coding systems across trial sites.

Our analysis has some potential limitations. First, the costing methods analyzed do not include non-facility costs, such as physician costs, office visits, or other ancillary services that are usually not captured in UB-04s. These types of HCU could be relevant for procedures where outpatient HCUs are frequently used. Second, as UB-04s are not collected uniformly across sites, our study doesn't account for potential differences in costs between HCUs observed in UB-04s and the costs in utilizations that occurred in sites that didn't collect UB-04s. Case reports will record these utilizations, but if UB-04s are not collected then it is not possible to compare costs calculated through cost assignments and CRFs. And lastly, the sample sizes are relatively small, particularly the sample of the WRAP-IT CIED infection cost analysis, which makes the results susceptible to outliers and parametric choices.

There are four approaches for assigning costs to health services alongside clinical trials, including microcosting, cost-adjusted charges, cost assignment process (also known as unit costing or standardized resource use assignment), and gross costing. These costing approaches focus on either a micro or an aggregate level and can be conducted using sources internal or

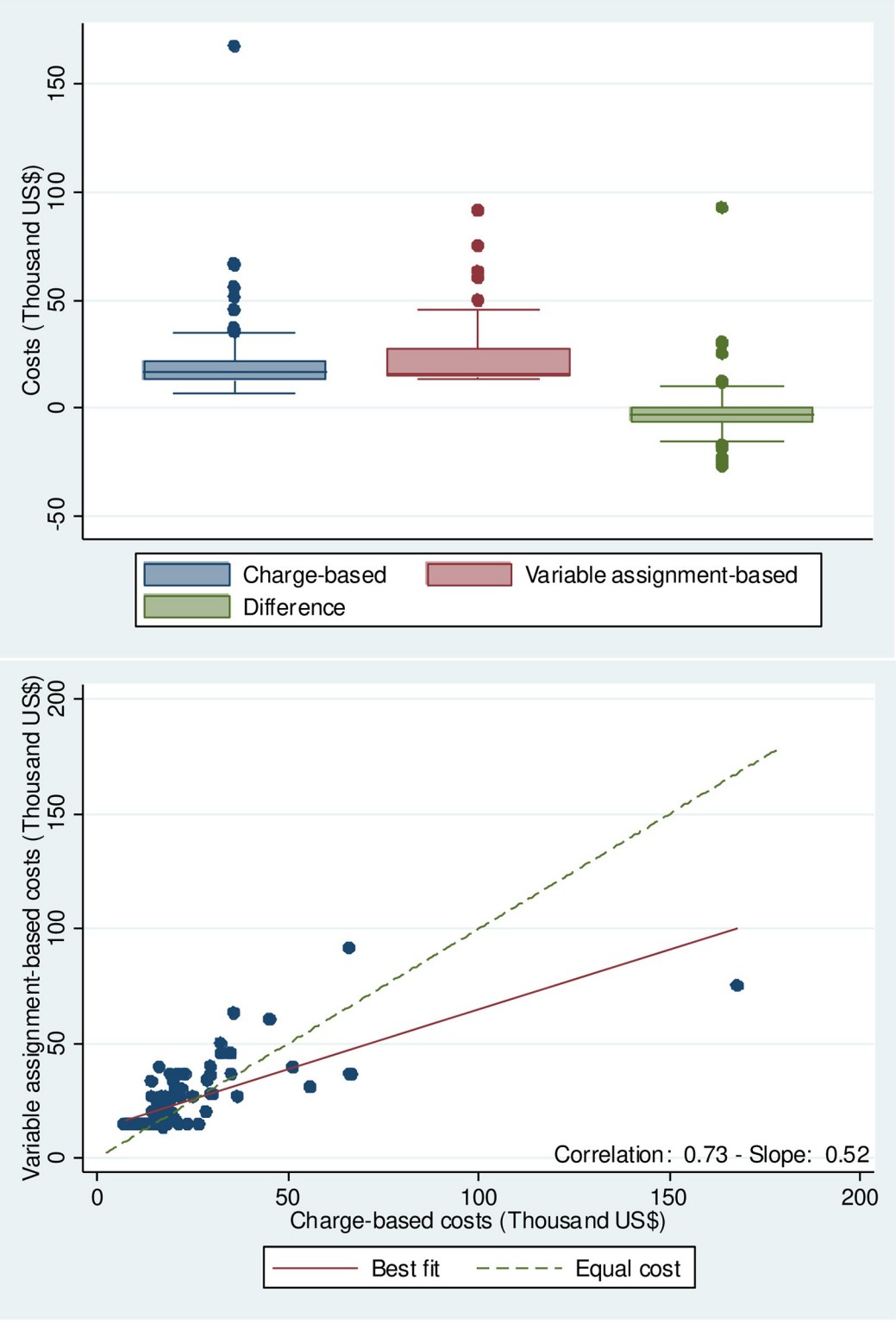

**Fig 3. Alternative charge-based and variable assignment-based costs for anomalous Micra IDE implants.** Results for the alternative Charge- and variable based costs using data from the 16/28 Micra IDE implant sites (57%) is shown. A) shows a box and whisker plot for cost (in thousand US $) with Charge-based costs are shown in blue, assignment-based costs are shown in red and the difference between the two in green. B) shows a scatterplot with the Charge-based costs (x-axis) vs the Assignment bases costs (y-axis) with the best fit line denoted with a solid red line and equal cost shown with a dotted green line.

external to the trial. Prior research suggests the choice between cost-determination methods represents a trade-off between precision and expense [9].

Medical bills provide an overall picture of patient encounters in the healthcare system and can provide accurate data on patient-level charges in a specific site. The cost-adjusted charges approach directly uses the medical bills of the study subjects enrolled in the trial to assign cost. One way to accomplish this is using UB-04 billing information to document patient hospital charges. The primary advantage of this approach is it provides a more accurate estimate of the cost that was incurred within the trial. In addition, the approach minimizes the need to collect data on resource utilization separately from unit costs for some cases [1].

However, there are several disadvantages to this approach. First, this is an expensive and time-consuming approach, since trial sponsors and hospital research staff need to spend extra time and resources to collect UB-04s and extract the data. Second, hospital charges can be twice as much as the actual cost of care, thus raw charges should not be used as an estimate of the cost of care [10]. This can be addressed by adjusting charges using CCRs, yet, caution is warranted since charges for a specific service may not be proportionate to economic cost [11]; in addition, the timing of the publication of CCRs is lagged by several years. Third, some researchers suggest costing to be more accurate if cost adjustment is performed at the department level instead of hospital level [10]. Finally, generalizability of this method is a significant issue since charges may be difficult to obtain for care received at other sites, UB-04s are only collected for some US sites, and the forms are never available for sites outside the US.

Alternatively, the cost assignment approach applies costs to each health care resource consumed by the patient[1] and requires only the healthcare utilization data to be collected by CRFs. In this approach the cost of a specific event is estimated as the sum of the costs for inpatient room and board, outpatient visit, emergency room visit, physician time, tests/procedures performed, and drugs received [12]. The unit costs are acquired from real world data for patients in clinical practice settings. There are several advantages in using the cost assignment process. First, this approach is regarded as more accurate than cost-adjusted charges. The cost assignment process reduces the variability in cost due to the fact that estimates are based on a wider selection of hospitals and are more representative from a broad healthcare system perspective [1]. In addition, this method eliminates the problems caused when billing information is missing for some centers or patients [9] and overcomes the additional burden on sponsors and sites to collect UB-04s. Second, the cost assignment process is easily replicable because it requires aggregate data instead of -site-specific information. One potential disadvantage of the cost assignment approach is that the real-world data used to calculate costs may not include the resources consumed in the trial or may not be representative of the whole population.

Our findings show that assigning costs to health services based on clinical information collected through CRFs achieve similar results to transforming charges from UB-04s into costs in the WRAP-IT CIED infections and Micra adverse event cases; the average costs obtained from either method were not statistically different from each other. For Micra implant costs, both methods estimate different costs; these differences did not disappear when refining the assignment-based method by correcting inaccuracies in UB-04 reporting. As Micra devices were recent technologies not widely adopted, finding matches in real-world data like the PHD was challenging. These lack of matches and possible inconsistencies by hospitals in their UB-04 reporting could cause the differences between costing methods. An assignment-based costing method is limited by the availability of real-world data that matches the health services in the clinical studies. For instance, the PHD includes hospital-based encounters only; for this study, outpatient utilizations not hospital-based were not included. On the other hand, charge-based methods are limited by the quality of the financial information in UB-04s. In the Micra implants case, some charge-based costs were less than the purchase price of the device, which

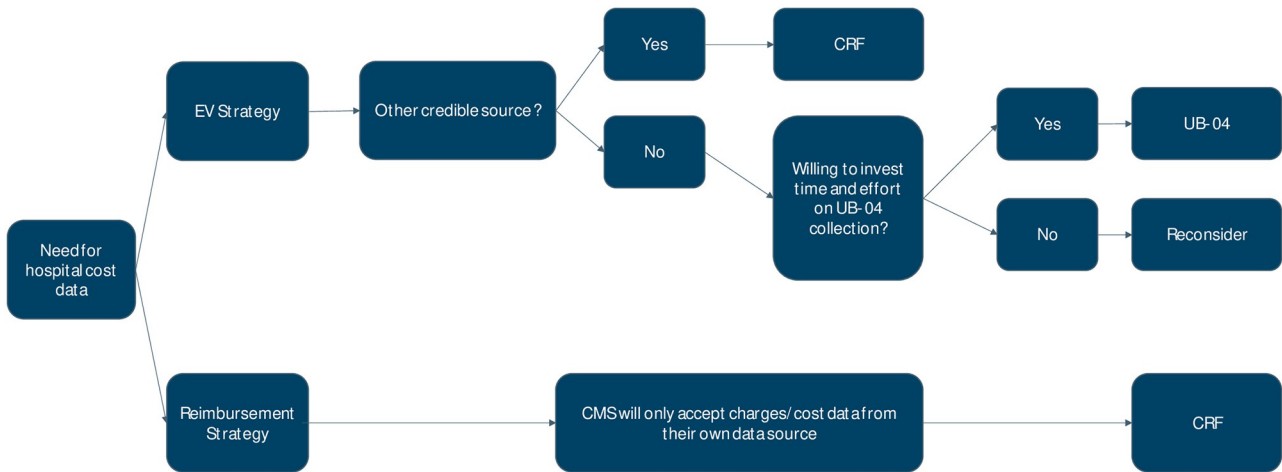

**Fig 4. Decision tree for CRF vs UB-04.** A decision tree showing the authors proposal on how to decide whether to use data from case report forms (CRFs) vs UB-04s.

casts doubt on the financial information in the UB-04s. In this case, neither assignment-based nor charge-based costing methods seem reliable, and perhaps a microcosting approach is more appropriate.

In the current environment where there is increased focused on value-based care, there is often a need for an economic analysis of clinical trial outcomes. To weigh the pros and cons of using either cost-adjusted charges or cost-assignment approaches for value analysis with real-world data, Barnett et al proposed an enhanced 14-point checklist [11]. The checklist considers the weakness of each method and is designed to help researchers avoid bias in assessing the cost of services most directly affected by the intervention under study. In addition to this checklist, and in consideration of the pros and cons of each method outlined above, we developed a decision tree based on our results describing when to collect economic data through CRFs vs UB-04s (Fig 4).

The needs for economic data collection during a clinical trial were clustered in two basic groups, first for the purpose of an economic value (EV) strategy and second for a reimbursement strategy. In the case of an EV strategy, the first thing to consider would be whether there is an alternative credible source to obtain the cost data instead of collecting UB-04s. Important things to consider while making this decision would be: i) whether the population or treatment of interest can be identified in a claims database; ii) whether healthcare utilization can be obtained; iii) if cost data can be extracted from a claims database; iv) the perspective of the analysis to be conducted (hospital, payer); v) would the analysis require encounter level or longitudinal data. Based on our experience, if there is no alternative credible source to obtain economic data and the research team is willing to invest adequate time and effort to fund and support the effort, then the suggested decision is to collect UB-04s. For any other answer the suggested decision is to use CRFs or reconsider the economic data collection strategy. In the case where economic data are needed for a reimbursement strategy, since CMS will only accept cost data from their own data source, the suggested decision would be always to collect the data through CRF.

Overall, the circumstances in which it would be useful to collect UB-04 information would be very narrow; however, it has been the best option in some cases. For instance, during the LATERAL trial Mokadam et al collected UB-04s to assess the cost of implantation via thoracotomy of a left ventricular assist device (LVAD) and compare that to the cost of implantation

via the standard implant technique of sternotomy [13]. This allowed the authors to circumvent the fact that there is no specific code to identify an LVAD implantation via thoracotomy in a claims database and still complete a value assessment. This case differs from the Micra implant case in that it studied the implantation of a known viable treatment option for advanced heart failure using a less invasive technique, while the Micra was a completely novel technology.

## Conclusions

This study compares two methods of assigning costs to clinical study data. We compare an assignment-based method that matched costs from real-world data to observed health services, to a charges-based method that transformed charges reported in UB-04 forms into costs. We find that cost assignment methods achieve results like charge-based costs, without the additional time and effort of collecting UB-04 forms. The assignment-based method is challenging to implement for services or technologies that are not mature in a health care system, but it is not clear if using UB-04 information is a better option in similar circumstances. Finally, for medical interventions or services where there is no credible source to estimate their costs, UB-04s are a viable option.

## Supporting information

**S1 Fig. UB-04 form.** Source: https://www.cdc.gov/wtc/pdfs/policies/ub-40-P.pdf.
(PDF)

**S1 Table. Procedure and diagnosis codes.** CIED: Cardiac implantable electronic device. CPT: Current procedural terminology. ICD-10-CM: International classification of diseases– 10th version–clinical modification. IDE: Investigational device exemption.
(XLSX)

**S1 Dataset.**
(ZIP)

## Acknowledgments

We are grateful to Karissa Alm for Case Report Forms evaluation and DRG assignment, Jiani Zhou and Argyro Azariadi for UB-04 data extraction, Patrick Zimmerman for statistical review, and Amy Molan for technical writing assistance.

## Author Contributions

**Conceptualization:** Eleni Ismyrloglou, Xiaoxiao Lu, Jennifer Hinnenthal, Reece Holbrook.

**Data curation:** Lucas Higuera.

**Formal analysis:** Lucas Higuera, Reece Holbrook.

**Investigation:** Eleni Ismyrloglou, Xiaoxiao Lu, Jennifer Hinnenthal, Reece Holbrook.

**Methodology:** Lucas Higuera, Jennifer Hinnenthal, Reece Holbrook.

**Project administration:** Jennifer Hinnenthal, Reece Holbrook.

**Resources:** Eleni Ismyrloglou, Jennifer Hinnenthal.

**Validation:** Reece Holbrook.

**Writing – original draft:** Lucas Higuera, Eleni Ismyrloglou, Xiaoxiao Lu, Jennifer Hinnenthal, Reece Holbrook.

**Writing – review & editing:** Lucas Higuera, Eleni Ismyrloglou, Xiaoxiao Lu, Jennifer Hinnenthal, Reece Holbrook.

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
