## [Decision Letter · Decision Letter 0]

4 Jul 2022

PONE-D-21-30170Collection of economic data using UB-04s: is it worth the effort? Evidence from two clinical trials.PLOS ONE

Dear Dr. Higuera,

Thank you for submitting your manuscript to PLOS ONE. After careful consideration, we feel that it has merit but does not fully meet PLOS ONE’s publication criteria as it currently stands. Therefore, we invite you to submit a revised version of the manuscript that addresses the points raised during the review process.

We look forward to receiving your revised manuscript.

Kind regards,

Vijay S. Gc, PhD

Academic Editor

PLOS ONE

Journal Requirements:

2. Thank you for stating the following in the Financial Disclosure section: "LH, EI, JH, and RH are all employees of Medtronic plc. XL is a former employee of Medtronic plc." 

We note that you received funding from a commercial source: [Name of Company]

Reviewers' comments:

Reviewer's Responses to Questions

**Comments to the Author**

1. Is the manuscript technically sound, and do the data support the conclusions?

Reviewer #1: Yes

2. Has the statistical analysis been performed appropriately and rigorously? 

Reviewer #1: Yes

3. Have the authors made all data underlying the findings in their manuscript fully available?

Reviewer #1: Yes

4. Is the manuscript presented in an intelligible fashion and written in standard English?

Reviewer #1: Yes

5. Review Comments to the Author

Reviewer #1: The current study compares a broad, charge-based costing methodology to a more detailed healthcare resource utilization (HCU) data assignment-based costing methodology across three different healthcare encounter types common to two Medtronic trials with available UB-04 (Medicare summary billing) data. Encounters for 23 CIED infections within the WRAP-IT trial, 66 IDE-related adverse events from the Micra IDE trial, and 108 Micra IDE implant encounters from the same trial were costed using (a) UB-04 total charges converted to costs using publicly available CCRs at a hospital-and-year level and (b) provider costs from Premier Health Data for encounters matched >=2:1 to those in the trials, with matching criteria dependent on the encounter type and place of service, among other variables. Differences in the estimated costs by costing methodology were tested statistically for difference from 0. The authors found that methodologies derived statistically similar costs for the adverse event and infection encounters. Costing methodologies led to statistically different costs among the implant encounters, and neither methodology was considered clearly superior: UB-04s underrepresented true costs on average, and the cost assignment strategy relied on matching trial cases to PHD claims generated one year after the Micra IDE was approved for use, when adoption was still relatively low. The authors appropriately conclude that their total charge UB-04-based and cost-assignment-based costing may both be inadequate in deriving costs for novel interventions without established coding and reimbursement strategies.

In order to guide decisions on whether to invest resources into collection of primary costing data alongside clinical trials, the investigators leveraged their findings to develop a decision tree for selecting costing strategies depending on purpose of the analysis (economic value or reimbursement strategy). The research article adds to the health economics methodological literature, provides a tangible resource for making important cost data collection decisions, and satisfies all criteria for publication in PLOS ONE. I recommend the article for publication after minor revisions.

Study methodology was transparent and succinctly described. The data sources for the different costing strategies are clearly explained, as are the statistical methods used to compare total encounter costs by strategy. Use of GLM modeling with gamma family is appropriate for cost comparisons, and inclusion of patient and site-patient random effects estimates reflects statistical rigor. No mention of the small sample size (23 bills in the WRAP-IT CIED infection comparison, in particular) is made in the limitations or the statistical methodology sections and should be considered.

There are a few additional minor issues the authors are suggested to address. (1) The paragraph beginning at line 219 is confusing. The authors are suggested to clarify that the total cost estimates that fell below the expected purchase price of the Micra device (line 220) are the charge-based costs rather than the assignment-based costs. This only became clear once it was mentioned in the Discussion section later. (2) Inflation of the costs to 2017 USD for the WRAP-IT infection data and to 2018 USD for the Micra data is unlikely to affect comparability of the cost differences between costing methodologies across the 3 encounter types, but it is unclear why the WRAP-IT cost data are not similarly inflated to 2018 to pointedly ensure this comparability, since total cost comparisons are differences rather than ratios. (3) Line 71-73 regarding the ISPOR Task Force Report on criteria for collecting economic data lacks information. Why are the available resources inadequate for decision making, and/or how will the original research study add important information to the resources for deciding whether to collect UB-04s or claims/administrative data alongside clinical trials? (4) An extra word in line 158 should be removed ("Since health care costs data {are} usually follow..."). (5) A typo in line 248 should be corrected ("These types of HCU could be relevant for procedures {were} outpatient HCUs are frequently used" -- "were" should be "where").

This type of direct comparison of costing methodologies is of high interest to health economists, especially those who engage in primary cost data collection. The study is thoughtfully executed and the article is well-written and -organized.

6. PLOS authors have the option to publish the peer review history of their article (what does this mean?). If published, this will include your full peer review and any attached files.

Reviewer #1: **Yes: **Katherine Robertus Vilain

---

## [Author Response · Author response to Decision Letter 0]

21 Sep 2022

Journal Requirements:

ANS: we have formatted the manuscript to follow PLOS ONE style requirements.

2. Thank you for stating the following in the Financial Disclosure section: "LH, EI, JH, and RH are all employees of Medtronic plc. XL is a former employee of Medtronic plc." 

We note that you received funding from a commercial source: [Name of Company]

ANS: we have updated our cover letter, adding a statement on the funding of the commercial source (Medtronic) and stating the role of the funder in the study. The following statement regarding funding and competing interests was added: "This study was funded by Medtronic, as well as the clinical trials used in it; this does not alter our adherence to PLOS ONE policies on sharing data and materials. The publication of study results was not contingent on Medtronic’s approval or censorship of the manuscript. All authors, except for Dr. Lu, currently receive wages and benefits from Medtronic, and are minority stockholders of Medtronic. Dr. Lu received wages and benefits from Medtronic as a former employee." 

ANS: we have provided minimal underlying data sets (3, one per cost study) in the submitted file 'Datasets.zip'.

Reviewers' comments:

Reviewer's Responses to Questions

Comments to the Author

5. Review Comments to the Author

Reviewer #1: The current study compares a broad, charge-based costing methodology to a more detailed healthcare resource utilization (HCU) data assignment-based costing methodology across three different healthcare encounter types common to two Medtronic trials with available UB-04 (Medicare summary billing) data. Encounters for 23 CIED infections within the WRAP-IT trial, 66 IDE-related adverse events from the Micra IDE trial, and 108 Micra IDE implant encounters from the same trial were costed using (a) UB-04 total charges converted to costs using publicly available CCRs at a hospital-and-year level and (b) provider costs from Premier Health Data for encounters matched >=2:1 to those in the trials, with matching criteria dependent on the encounter type and place of service, among other variables. Differences in the estimated costs by costing methodology were tested statistically for difference from 0. The authors found that methodologies derived statistically similar costs for the adverse event and infection encounters. Costing methodologies led to statistically different costs among the implant encounters, and neither methodology was considered clearly superior: UB-04s underrepresented true costs on average, and the cost assignment strategy relied on matching trial cases to PHD claims generated one year after the Micra IDE was approved for use, when adoption was still relatively low. The authors appropriately conclude that their total charge UB-04-based and cost-assignment-based costing may both be inadequate in deriving costs for novel interventions without established coding and reimbursement strategies.

In order to guide decisions on whether to invest resources into collection of primary costing data alongside clinical trials, the investigators leveraged their findings to develop a decision tree for selecting costing strategies depending on purpose of the analysis (economic value or reimbursement strategy). The research article adds to the health economics methodological literature, provides a tangible resource for making important cost data collection decisions, and satisfies all criteria for publication in PLOS ONE. I recommend the article for publication after minor revisions.

Study methodology was transparent and succinctly described. The data sources for the different costing strategies are clearly explained, as are the statistical methods used to compare total encounter costs by strategy. Use of GLM modeling with gamma family is appropriate for cost comparisons, and inclusion of patient and site-patient random effects estimates reflects statistical rigor. No mention of the small sample size (23 bills in the WRAP-IT CIED infection comparison, in particular) is made in the limitations or the statistical methodology sections and should be considered.

ANS: We added a sentence on this study limitation to the discussion section: “And lastly, the sample sizes are relatively small, particularly the sample of the WRAP-IT CIED infection cost analysis, which makes the results susceptible to outliers and parametric choices.” 

There are a few additional minor issues the authors are suggested to address. (1) The paragraph beginning at line 219 is confusing. The authors are suggested to clarify that the total cost estimates that fell below the expected purchase price of the Micra device (line 220) are the charge-based costs rather than the assignment-based costs. This only became clear once it was mentioned in the Discussion section later. (2) Inflation of the costs to 2017 USD for the WRAP-IT infection data and to 2018 USD for the Micra data is unlikely to affect comparability of the cost differences between costing methodologies across the 3 encounter types, but it is unclear why the WRAP-IT cost data are not similarly inflated to 2018 to pointedly ensure this comparability, since total cost comparisons are differences rather than ratios. (3) Line 71-73 regarding the ISPOR Task Force Report on criteria for collecting economic data lacks information. Why are the available resources inadequate for decision making, and/or how will the original research study add important information to the resources for deciding whether to collect UB-04s or claims/administrative data alongside clinical trials? (4) An extra word in line 158 should be removed ("Since health care costs data {are} usually follow..."). (5) A typo in line 248 should be corrected ("These types of HCU could be relevant for procedures {were} outpatient HCUs are frequently used" -- "were" should be "where").

ANS: Thank you for your comments. Here are our detailed answers:

(1) We have clarified that indeed we refer to the charge-based costs being above the price of a Micra device. 

(2) We adjusted costs in the WRAP-IT CIED infection analysis to 2017 US dollars instead of 2018 US dollars to be consistent with previous economic analysis of the WRAP-IT trial (Wilkoff B et al., 2020). As the reviewer noted, the year chosen to adjust costs does not affect comparability within each cost study. We have added the reason why we adjusted WRAP-IT CIED infection costs to 2017 US dollars instead of 2018 US dollars. 

(3) The reviewer is correct that the reference to the ISPOR report lacks information. We have added a sentence stating what the report recommends, and what does it lack that this study provides: “This report states the possible trade-offs across accuracy, feasibility, generalizability, and cost among costing methods, but the report doesn’t provide specific comparisons across costing methods.”

(4) and (5) Thank you reviewer for your keen eye. We have corrected both problems.

---

## [Editor Report · Decision Letter 1]

2 Nov 2022

Collection of economic data using UB-04s: is it worth the effort? Evidence from two clinical trials.

PONE-D-21-30170R1

Dear Dr. Higuera,

We’re pleased to inform you that your manuscript has been judged scientifically suitable for publication and will be formally accepted for publication once it meets all outstanding technical requirements.

Kind regards,

Vijay S. Gc, PhD

Academic Editor

PLOS ONE
---

## [Editor Report · Acceptance letter]

8 Nov 2022

PONE-D-21-30170R1 

Collection of economic data using UB-04s: is it worth the effort? Evidence from two clinical trials. 

Dear Dr. Higuera:

I'm pleased to inform you that your manuscript has been deemed suitable for publication in PLOS ONE. Congratulations! Your manuscript is now with our production department. 

Kind regards, 

on behalf of

Dr. Vijay S. Gc 

Academic Editor

PLOS ONE